# Assessment of front and back of pack nutrition labels of selected convenience food products and snacks available in the Indian market

Shanmugam Shobana[1]*, Gopalakrishnan Sangavi[1], Ramatu Wuni[2], Bakshi Priyanka[1], Arun Leelavady[1], Dhanushkodi Kayalvizhi[1], Ranjit Mohan Anjana[3], Kamala Krishnaswamy[4], Karani Santhanakrishnan Vimaleswaran[2,5]*, Viswanathan Mohan[3]

1 Department of Diabetes Food Technology, Madras Diabetes Research Foundation, Chennai, Tamil Nadu, India, 2 Hugh Sinclair Unit of Human Nutrition, Department of Food and Nutritional Sciences and Institute for Cardiovascular and Metabolic Research (ICMR), University of Reading, Reading, United Kingdom, 3 Department of Diabetology, Madras Diabetes Research Foundation, Chennai, Tamil Nadu, India, 4 National Institute of Nutrition, Hyderabad, India, 5 Institute for Food, Nutrition and Health (IFNH), University of Reading, Reading, United Kingdom

* shobanashanmugam@mdrf.in (SS); v.karani@reading.ac.uk (KSV)

## Abstract

Nutrition labels on packaged food items provide at-a-glance information about the nutritional composition of the food, serving as a quick guide for consumers to assess the quality of food products. The aim of the current study is to evaluate the nutritional information on the front and back of pack labels of selected packaged foods in the Indian market. A total of 432 food products in six categories (*idli* mix, breakfast cereals, porridge mix, soup mix, beverage mix and extruded snacks) were investigated by a survey. Nutritional profiling of the foods was done based on the Food Safety and Standards Authority of India (FSSAI) claims regulations. The healthiness of the packaged foods was assessed utilising nutritional traffic light system. The products were classified into 'healthy', 'moderately healthy' and 'less healthy' based on the fat, saturated fat, and sugar content. Most of the food products evaluated belong to healthy' and 'moderately healthy' categories except for products in extruded snacks. Reformulation of 'extruded snacks' are necessary to decrease the total and saturated fat content. The nutrient content claims were classified using the International Network for Food and Obesity / NCDs Research, Monitoring and Action Support (INFORMAS) taxonomy. Protein, dietary fibre, fat, sugar, vitamins and minerals were the most referred nutrients in the nutrient content claims. Breakfast cereal carried highest number of nutritional claims while porridge mix had the lowest number of claims. The overall compliance of the nutrient content claims for the studied food products is 80.5%. This study gives an overall view about the nutritional quality of the studied convenience food products and snacks in Indian market.

**Data Availability Statement:** The data analyzed in the manuscript was collected from front and back of pack nutritional labels of various food products mentioned in the manuscript. The collected data was transcribed into a spreadsheet and used for analysis. All relevant data are within the manuscript and its Supporting Information files.

**Funding:** 1. The Principal Investigator of the study (Dr.SS) from the Madras Diabetes Research Foundation acknowledge the funding received from the Department of Science and Technology (https://dst.gov.in/) – Technological Interventions for the Disabled and Elderly (DST – TIDE Scheme – Grant No: SEED/TIDE/2018/63) for conducting this study. The funder didn't play any role in the study design, data collection and analysis, decision to publish, or preparation of the manuscript. 2. Dr. KSV acknowledges the support from Medical Research Council (grant # Grant no.MR/S024778/1).

**Competing interests:** The authors have declared that no competing interests exist.

## Introduction

The demand for convenience foods is one of the major trends in the food industry globally [1, 2]. The convenience foods industry is forecast to grow at a compound annual growth rate (CAGR) of 4.3% by 2025 [1], and according to Statista [3] the revenue generated by the sector in India is around 58 billion US dollars with a predicted CAGR of 9.55% between 2022 and 2027. It has been suggested that the growing demand is driven by urbanisation and lifestyle changes including the rise in nuclear families; busy schedules such as long working hours and commuting distances; changing taste preferences; and a tendency to free up time for leisure activities [1, 2, 4]. The types of convenience foods available in the market in India include ready-to-eat (RTE) foods, such as breakfast cereal bars, which do not require heating and ready-to-cook foods (RTC) such as instant curried dhal mix which requires heating [1]. Nevertheless, the demand for convenience foods is a growing concern over health benefits, given the rising prevalence of obesity, type 2 diabetes (T2D) and other cardio-metabolic diseases in India [5, 6]. The rise in non-communicable diseases (NCDs) has been attributed to a nutrition transition characterised by increased consumption of processed foods rich in calories, saturated fat, simple carbohydrates and sodium [6, 7]. In order to make healthy choices, nutrition labels have a role to play as they provide information on the nutritional composition of food, serving as a quick guide for consumers to assess the quality of food products [2, 8]. According to a systematic review of 120 studies [8], nutrition labels are viewed as a highly trusted source of information and many consumers rely on them in the selection of food products. Nonetheless, nutrition labelling in India is at an evolving stage and evidence of awareness and understanding of these labels by the Indian population is limited [2, 9]. In 2011, food safety and standards (packaging and labelling) regulations were introduced by the Food Safety and Standards Authority of India (FSSAI) [10] and manufacturers of pre-packaged foods are required to follow these guidelines. In the context of increasing prevalence of T2D and other cardiometabolic diseases in India [7, 11–13] few studies have examined the compliance of packaged foods to nutrition guidelines [1, 2]. Hence, the objective of the present study was to evaluate the nutritional profile and claims of selected convenience food products and snacks in the Indian market. In addition to that, the study also measured the healthiness of the foods by categorising them according to their nutrient profiles.

## Methodology

### Data collection

A physical (in Chennai, India) and online market survey was conducted to collect data on the front-of-pack (FOP) and back-of-pack (BOP) of selected convenience foods sold in India. For the online survey, we chose the home shopping site because of the opportunity to access a wider range of products. The products available on the site were similar to those available in a physical supermarket. The overview of the market survey protocol is shown in Fig 1. The current study includes data from 432 convenience food products comprising five categories (*idli* mix, breakfast cereals, porridge mix, soup mix, beverage mix), and extruded snacks available under different brand names. For each product, the following data were collected and recorded in an Excel spreadsheet: brand name, product name, energy (kcal), protein (g/100 g), fat (g/100 g), saturated fat (SFA) (g/100 g), trans fat (mg/100 g), cholesterol (mg/100 g), carbohydrate (g/100 g), sugar (g/100 g), fibre content (g/100 g), sodium content (mg/100g), ingredients list, and nutrition content claims. In a few products, the nutrient content was reported as <1g/<0.1g and the data was entered as 0.5g/0.05g respectively in the spreadsheet. The salt content (g) was converted to sodium content (mg) by multiplying by a conversion factor of 400 [14].

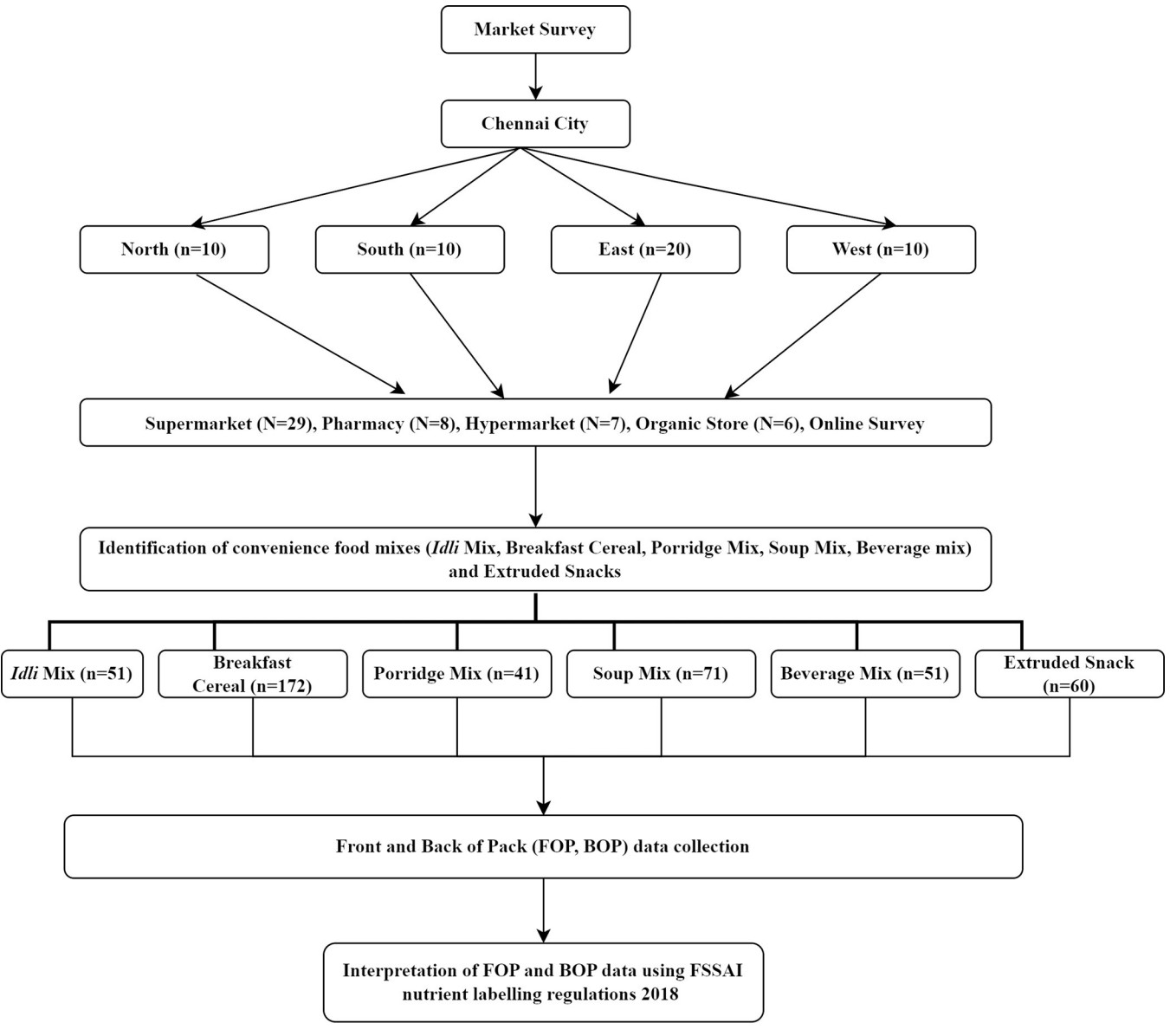

**Fig 1. Market survey–Study protocol.**

The nutritional quality of the products was assessed based on the ingredient composition and the nutritional values (for 100g) declared on the food label.

## Data analysis

Primarily, we analysed the descriptive statistics (mean, standard deviation, range) of the nutrient content of convenience foods. The percentage energy distribution was computed using the mean values of carbohydrates, protein and fat for the food products. The categorization of food products according to their nutrient content with implications on public health is termed as 'nutritional profiling' [15]. In this study, nutritional profiling of convenience food products was performed according to the criteria of FSSAI advertising and claim regulations, 2018 [16]. In addition to that, we have categorised the food products using the traffic light system

**Table 1. Criteria for nutritional profiling of foods (per 100g of solids).** (Source: FSSAI, 2018).

| S. No | Nutrition claim | Condition |
|---|---|---|
| 1 | Low fat | The product contains not more than 3 g of fat |
| 2 | Fat-free | The product contains not more than 0.5 g of fat |
| 3 | Low cholesterol | The product contains not more than 20 mg of cholesterol & 1.5 g of saturated fat |
| 4 | Cholesterol free | The product contains not more than 5 mg of cholesterol. Additionally, the food shall contain no more than 1.5 g SFA |
| 5 | Low SFA | The product contains not more than 1.5 g of SFA |
| 6 | SFA free | The SFA content of the product does not exceed 0.1 g |
| 7 | Trans-fat-free | The product contains less than 0.2g trans-fat |
| 8 | Low sugar | The product contains not more than 5 g of sugars |
| 9 | Sugar-free | The product contains not more than 0.5 g of sugars |
| 10 | Protein [Source] | The product covers 10% of Recommended Dietary Allowance (RDA) |
| 11 | Protein [Rich/High] | The product covers 20% of RDA |
| 12 | Dietary fibre [Source] | The product contains at least 3 g of fibre |
| 13 | Dietary fibre [Rich/High] | The product contains at least 6 g of fibre |
| 14 | Sodium [Low] | The product contains not more than 0.12 g/120mg of sodium per 100 g for solids |
| 15 | Sodium [Very low] | The product contains not more than 0.04 g/40mg of sodium per 100 g for solids |
| 16 | Sodium [Free] | The product contains not more than 0.005g/5 mg of sodium per 100 g for solids |
| 17 | Glycemic index—GI [Low GI] | GI value below 55 |

Source: Food Safety and Standards (Advertising and Claims) Regulations, 2018. Available: https://fssai.gov.in/upload/uploadfiles/files/Compendium_Advertising_Claims_Regulations_04_03_2021.pdf

Recommended Dietary Allowances (RDAs) are the levels of intake of essential nutrients that, on the basis of scientific knowledge, are judged by the Food and Nutrition Board to be adequate to meet the known nutrient needs of practically all healthy persons.

Glycemic index (GI): A food's GI indicates the rate at which the carbohydrate in the food is broken down into glucose and absorbed from the gut into the blood and expressed as a percent of the response to the same amount of carbohydrate from a standard food.

recommended by the UK government. The traffic light system categorizes the key nutrients in foods (total fat, saturated fat and total sugar) as low, medium and high respectively. The low (healthy), medium (moderately healthy) and high (least healthy) categories are represented with green, amber and red colours respectively [17]. The criteria for nutritional profiling of foods are given in Tables 1 and 2. Both FSSAI and traffic light system provided nutrient thresholds for 100g of the product. In addition to that, the first/main ingredients of convenience food products were studied. A representation that implies a food product has certain qualities

**Table 2. Criteria for nutritional profiling of foods (per 100g of solids).** (Source: FSA, 2016).

| Food composition | Condition for the categories | | |
|---|---|---|---|
| | Low | Medium | High |
| Total fat | $\leq 3.0$ g | > 3.0 g to $\leq 17.5$ g | > 17.5 g |
| Saturated fat | $\leq 1.5$ g | > 1.5 g to $\leq 5.0$ g | > 5.0 g |
| Total sugar | $\leq 5.0$ g | > 5.0 g to $\leq 22.5$ g | > 22.5 g |

Source: FSA. Guide to creating a front of pack (FoP) nutrition label for pre-packed products sold through retail outlets. 2016. Available: https://www.gov.uk/government/publication

relating to its origin, nature, nutritional qualities, manufacture, processing, or any other quality is referred to as a claim [18]. The nutrition content claims describe the level of nutrients contained in the food product. The nutrition content claims were classified using the FSSAI advertising and claim regulations, 2018 [16]. In the present study, the compliance of the nutrient content claims of the food products was analysed.

## Results

### Nutritional information and profiling

The descriptive statistics information of the nutrients is shown in Table 3. All convenience food products were rich in carbohydrate content and beverage mixes were found to have the highest carbohydrate content (35.5g to 95g/100g). The beverage mixes were found to be rich in protein content (mean 15.8±8.1g/100g) followed by the *idli* mixes (mean 12.2±4.0g/100g). Extruded snacks had the high mean dietary fibre content (8.6±5.5g/100g), highest total fat content (mean 28.3±7.5 g/100g), SFA content (mean 11.0±4.5 g/100g) and energy (mean 536.1 ±69.9 kcal/100g). The soup mixes had high sodium levels (mean 3346.4±2228.0 mg/100g) and also showed the presence of traces of trans-fat (mean 0.2±0.9 mg/100g). The highest

**Table 3. Nutrient content (per 100g) declared on the pack of market foods.**

| Food Category | | Carbohydrate (g) | Protein (g) | Dietary fibre (g) | Fat | | | | Sodium (mg) | Sugar (g) | Energy (kcal) |
|---|---|---|---|---|---|---|---|---|---|---|---|
| | | | | | Total fat (g) | SFA (g) | Trans-fat (mg) | Cholesterol (mg) | | | |
| *Idli* mix (N = 51) | 'n' | 51 | 51 | 33 | 51 | 30 | 29 | 25 | 26 | 34 | 51 |
| | Range | 20–79.7 | 5.9–24 | 0.5–14 | 0.1–51.5 | 0.0–5.4 | 0.0–1.0 | 0.0–0.0 | 0.0–1500 | 0.0–7.0 | 30–626 |
| | Mean ±SD | 68.3±9.8 | 12.2±4.0 | 4.8±3.1 | 5.7±7.4 | 1.9±1.6 | 0.0±0.2 | 0.0±0.0 | 683.1 ±569.3 | 1.3±1.5 | 367.3±68.1 |
| Breakfast cereals (N = 172) | 'n' | 172 | 169 | 159 | 172 | 150 | 148 | 99 | 140 | 150 | 171 |
| | Range | 32–90 | 4–27.0 | 0.8–29.9 | 0–32 | 0.0–17 | 0.0–0.1 | 0.0–0.5 | 0.0–2510 | 0.0–36 | 282–596.7 |
| | Mean ±SD | 72.2±10.2 | 10.5±3.9 | 8.3±4.3 | 8.6±6.5 | 2.5±3.1 | 0.0±0.0 | 0.0±0.1 | 278.1 ±392.2 | 14.2 ±9.0 | 405.0±46.2 |
| Porridge mix (N = 41) | 'n' | 41 | 41 | 40 | 36 | 8 | 12 | 10 | 26 | 23 | 41 |
| | Range | 25.2–90 | 3–24.9 | 1.0–12.2 | 0.3–13 | 0–1 | 0.0–0.0 | 0.0–0.0 | 0.01–296.2 | 0.0–42.5 | 115.5–525 |
| | Mean ±SD | 73.5±11.6 | 10.8±4.5 | 5.1±3.0 | 4.1±3.0 | 0.6±0.4 | 0.0±0.0 | 0.0±0.0 | 36.6±70.5 | 5.6±9.2 | 386±52.8 |
| Soup mix (N = 71) | 'n' | 71 | 71 | 55 | 66 | 65 | 53 | 48 | 59 | 62 | 71 |
| | Range | 4.5–78.0 | 0.4–51.1 | 0.2–13.8 | 0.5–19.2 | 0.0–6.9 | 0.0–5.0 | 0.0–49.0 | 4.4–8760.0 | 0.0–52.0 | 35.0–451.0 |
| | Mean ±SD | 54.4±23.6 | 9.6±9.9 | 4.3±3.7 | 4.3±3.6 | 1.6±1.3 | 0.2±0.9 | 2.7±8.8 | 3346.4 ±2228.0 | 16.6 ±14.8 | 298.2 ±111.3 |
| Beverage mix (N = 35) | 'n' | 35 | 35 | 28 | 35 | 28 | 31 | 23 | 32 | 33 | 35 |
| | Range | 35.5–95 | 2.2–34 | 0–22 | 0.3–20 | 0.2–9.1 | 0.0–0.1 | 0–35 | 25–668 | 0–83.7 | 351–455 |
| | Mean ±SD | 68.6±15.8 | 15.8±8.1 | 5.0±5.1 | 6.3±6.8 | 2.7±2.9 | 0.0±0.0 | 4.7±10.1 | 334.6 ±174.9 | 31.7 ±19.9 | 392.1±28.1 |
| Extruded snacks (N = 60) | 'n' | 60 | 60 | 20 | 60 | 54 | 52 | 22 | 47 | 59 | 60 |
| | Range | 39–78 | 3.6–17.5 | 2.1–18.0 | 8.9–45 | 2.2–18.9 | 0–0.19 | 0–2.5 | 175–1571 | 0–11.6 | 428–963 |
| | Mean ±SD | 60.0±8.1 | 7.4±2.1 | 8.6±5.5 | 28.3±7.5 | 11.0 ±4.5 | 0.1±0.1 | 0.1±0.5 | 781.8 ±208.9 | 3.3±2.5 | 536.1±69.9 |

N–Total number of convenience food products; n–Number of items reporting the parameter on the pack

**Table 4. Categorization of foods according to FSSAI regulations [16].**

| S.No | Nutrient claim | Percentage of food products categorized under the nutrient claim | | | | | |
|---|---|---|---|---|---|---|---|
| | | *Idly* mix | Breakfast cereals | Porridge mix | Soup mix | Beverage mix | Extruded snacks |
| 1 | Low fat | 37.3 | 22.1 | 31.7 | 44.4 | 54.3 | 0 |
| 2 | Fat-free | 3.9 | 1.2 | 2.4 | 0.0 | 2.9 | 0 |
| 3 | Low cholesterol | 49.0 | 57.6 | 24.4 | 63.9 | 60.0 | 36.7 |
| 4 | Cholesterol free | 49.0 | 57.6 | 24.4 | 58.3 | 48.6 | 36.7 |
| 5 | Low SFA | 27.5 | 48.8 | 19.5 | 52.8 | 37.1 | 0 |
| 6 | SFA free | 5.9 | 2.3 | 2.4 | 4.2 | 0.0 | 0 |
| 7 | Trans-fat free | 54.9 | 86 | 29.3 | 69.4 | 88.6 | 86.7 |
| 8 | Low sugar | 64.7 | 15.7 | 41.5 | 31.9 | 8.6 | 76.7 |
| 9 | Sugar-free | 17.6 | 7.6 | 7.3 | 12.5 | 5.7 | 11.7 |
| 10 | Protein [Source] | 100.0 | 96.5 | 95.1 | 56.9 | 97.1 | 93.3 |
| 11 | Protein [Rich/High] | 54.9 | 36.0 | 43.9 | 26.4 | 65.7 | 8.3 |
| 12 | Dietary fibre [Source] | 45.1 | 87.2 | 75.6 | 43.1 | 48.6 | 26.7 |
| 13 | Dietary fibre [Rich/High] | 19.6 | 68.6 | 36.6 | 20.8 | 28.6 | 21.7 |
| 14 | Sodium [Low] | 54.9 | 29.7 | 56.1 | 26.4 | 5.7 | 0.0 |
| 15 | Sodium [Very low] | 19.6 | 12.8 | 48.8 | 2.8 | 2.9 | 0.0 |
| 16 | Sodium [Free] | 17.6 | 4.1 | 29.3 | 2.8 | 0.0 | 0.0 |
| 17 | Glycemic Index [Low GI] | 3.9 | - | - | 1.4 | 11.4 | - |

Percentage of food products complying with the nutrient claim = [Number of food products complying with the claim/Total number of food products (N)] * 100

Source: Food Safety and Standards (Advertising and Claims) Regulations, 2018. Available: https://fssai.gov.in/upload/uploadfiles/files/Compendium_Advertising_Claims_Regulations_04_03_2021.pdf

cholesterol (49 mg/100g) was reported in the soup mix. Soup mixes also had poor protein [Mean 9.6 g/100g] and dietary fibre content [Mean 4.3g/100g]. Beverage mixes were found to be rich in sugar content (31.7±19.9g/100g).

The percentage of food products complying with the nutrient claims according to FSSAI is shown in Table 4. All the *idli* mixes available in the market were found to be a good source of protein and 54.9% of *idli* mixes are rich in protein. *Idli* mixes also had the highest percentage of products in the 'sugar-free' (17.6%) category. Except for soup mix, around 93–97% of food products in breakfast cereal, porridge mix, beverage mixes and extruded snacks are also found to be the source of protein. Beverage mix was found to have the highest percentage (65.7%) of products in the protein-rich category. In the case of dietary fibre, 87.2% of breakfast cereals are found to be a source and 68.6% are rich in dietary fibre. Around 50% of other convenience food products were found to be the source of dietary fibre excluding extruded snacks. Almost 87% - 89% of products in beverage mix, breakfast cereals and extruded snacks category were found to be trans-fat free. Though the highest cholesterol was reported in the soup mixes category, around 58.3% of soup mixes available in the market are cholesterol free. Almost 50% of the convenience food products, except for porridge mix (zero cholesterol) were found to be 'low cholesterol'. Most of the beverage mixes (54.3%) have a low-fat content, but *idli* mixes have the highest percentage of fat-free food products (3.9%). Porridge mixes were found to have the highest percentage of products in low sodium (56.1%), very low sodium (48.8%) and sodium-free (29.3%) categories. None of the products in extruded snacks fall under the low sodium category.

The categorization of convenience foods according to the nutritional traffic light system is shown in Fig 2. The traffic light graphs show that beverage mix has a maximum number of products in the high sugar category (65.7%) followed by soup mix (31%) and breakfast cereals

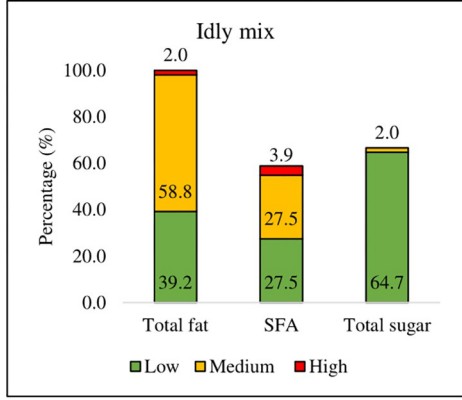

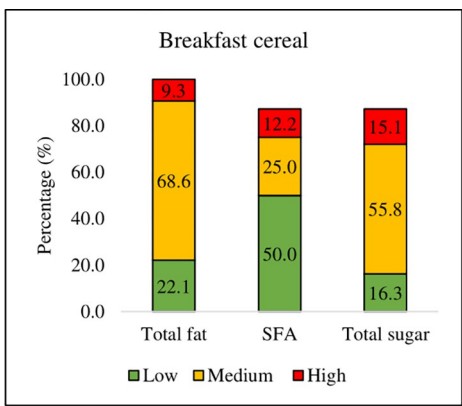

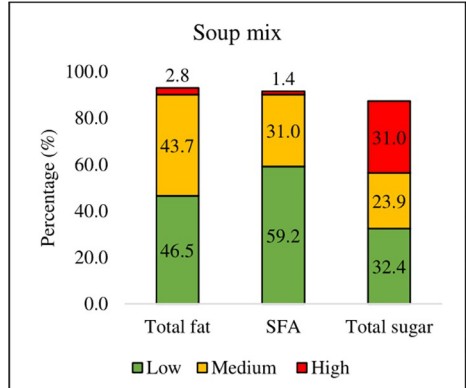

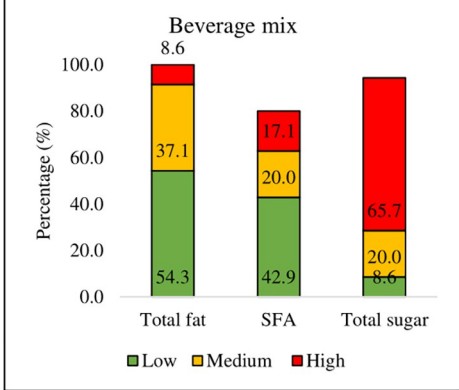

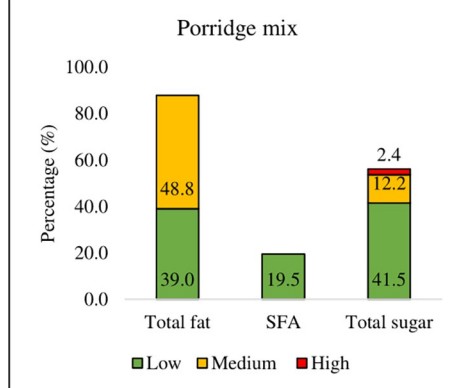

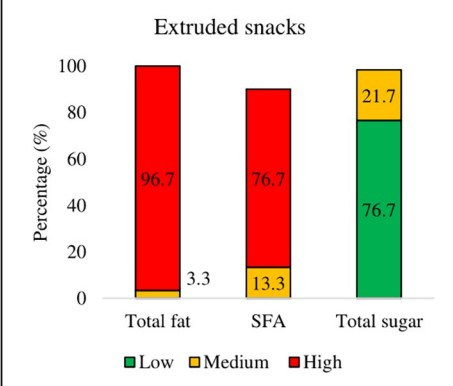

**Fig 2. Categorization of convenience foods and snacks according to nutritional traffic light system.**

(15.1%). *Idly* Mix has no products in the high sugar category. Likewise, extruded snack was found to have a maximum number of products in the high fat (96.7%) and high SFA (76.7%) categories. The porridge mix had no products in the high fat and high SFA categories. In the high fat category, soup mix and *idli* mix have 2.8% and 2.0% products respectively.

The mean percentage of energy contributed from carbohydrates, protein and fat for convenience foods is shown in S1 and S2 Figs and the distribution of nutrients among the market products is shown in S3 Fig. The graphs show that for all the foods (except extruded snacks), around 70% of the energy is contributed from carbohydrates. The energy distribution from protein is less than 15%. For extruded snacks highest energy contribution is from fats.

**Table 5. Percentage of main/first ingredient in convenience food products.**

| S. No | Ingredient Category | Ingredients included in the category | Percentage of main/first ingredient | | | | | |
|---|---|---|---|---|---|---|---|---|
| | | | *Idly* mix | Breakfast cereal | Porridge mix | Soup mix | Beverage mix | Extruded snacks |
| 1 | Millets | Barnyard millet, Finger millet, Pearl millet, Kodo millet, Little millet, Foxtail millet, Sorghum as well as millet flours and flakes (Pearl millet and sorghum flakes) | 25.5 | 11.0 | 51.2 | 0.0 | 14.3 | 0.0 |
| 2 | Rice | Rice, Red Rice, Mapillai Samba, Hand-pounded Red Rice, Sprouted Hand Pound Brown Rice and Rice flour, rice flour, rice grits and rice meal | 21.6 | 0.0 | 14.6 | 0.0 | 0.0 | 23.3 |
| 3 | Wheat | Wheat, Whole Wheat, Processed whole wheat, Broken wheat, Whole wheat flour (Atta), Refined Wheat Flour (Maida) as well as Wheat grits & flakes (Wheat flakes, Malted wheat flakes, Whole wheat flakes). | 2.0 | 20.9 | 17.1 | 9.7 | 11.4 | 1.7 |
| 4 | Barley | Barley | 0.0 | 0.0 | 2.4 | 0.0 | 22.9 | 0.0 |
| 5 | Quinoa | Quinoa, Quinoa flour and Extruded Quinoa Flakes | 2.0 | 3.5 | 2.4 | 0.0 | 0.0 | 0.0 |
| 6 | Maize/Corn | Corn, Corn Flakes, Corn Flour, Corn Grits and Corn meal | 0.0 | 11.6 | 4.9 | 8.3 | 0.0 | 46.7 |
| 7 | Pulses | Urad dhal, Green Moong Dhal, Black Gram, Green gram, Chana dal, Chickpeas, Red Massor, Whole Moong bean powder, Bengal gram powder | 7.8 | 0.0 | 7.3 | 4.2 | 0.0 | 1.7 |
| 8 | Semolina | Semolina, Wheat semolina, Little millet rava, Sorghum semolina, Finger millet semolina, Rice Semolina, Durum Wheat Semolina | 39.2 | 0.0 | 0.0 | 1.4 | 0.0 | 0.0 |
| 9 | Oats | Oats, rolled oats, Oats flakes, Instant oats, Whole rolled oats | 2.0 | 49.4 | 0.0 | 0.0 | 0.0 | 0.0 |
| 10 | Starch | Edible starch, Edible Potato Starch, Maize Starch, Mushroom Starch | 0.0 | 0.0 | 0.0 | 36.1 | 2.9 | 0.0 |
| 11 | Protein | Pea Protein, Whey protein, Whey protein caseinates, Texturized soy protein, Soy Protein Isolate, Peanut protein hydrolysate | 0.0 | 3.5 | 0.0 | 6.9 | 5.7 | 0.0 |
| 12 | Sugar | Sugar, Sucrose, Glucose syrup, Maltodextrin | 0.0 | 0.0 | 0.0 | 11.1 | 22.9 | 0.0 |
| 13 | Vegetables | Tomato, Asparagus, Cabbage, Potato, Sweet Potato, Carrot, Spinach leaves, Lemon, tepary beans flour | 0.0 | 0.0 | 0.0 | 19.4 | 0.0 | 18.3 |
| 14 | Herbs | Balloon vines, *Solanum trilobatum* | 0.0 | 0.0 | 0.0 | 2.8 | 0.0 | 0.0 |
| 15 | Milk powder | Milk solids, skim milk powder | 0.0 | 0.0 | 0.0 | 0.0 | 20.0 | 0.0 |
| 16 | Edible vegetable oil | Palmolein oil | 0.0 | 0.0 | 0.0 | 0.0 | 0.0 | 1.7 |

## Ingredient list

From the ingredients list, the main/first ingredient of the food products were collected and similar ingredients were aggregated into categories. The percentage of the main ingredients present in convenience foods is summarized in Table 5. As shown, the main ingredient in all convenience food products is cereals except for beverage mix. The percentage of the first ingredient in convenience food products are—*idly* mix (semolina– 39.2%), breakfast cereals (oats– 49.4%), porridge mix (millets– 51.2%), soup mix (starch– 36.1%), beverage mix (barley– 22.9% and sugar– 22.9%) and extruded snacks (corn– 46.7%) respectively.

## Nutritional content claims

The most recurrent nutritional content claims used in food products are source of protein, source of fibre, rich in protein or high protein, rich in fibre or high fibre, low sugar, sugar-free, trans-fat-free and cholesterol free. We have only evaluated the nutritional content claims related to protein, dietary fibre, fat, sugar and cholesterol in this study. The number of products displaying the nutritional content claims and the number of products complying with the nutritional content claims for each convenience food product is presented in Fig 3. The 'breakfast cereal' was found to have the highest number of nutritional content claims followed by the 'soup mix'. Beverage mixes had the least number of nutritional content claims. None of the

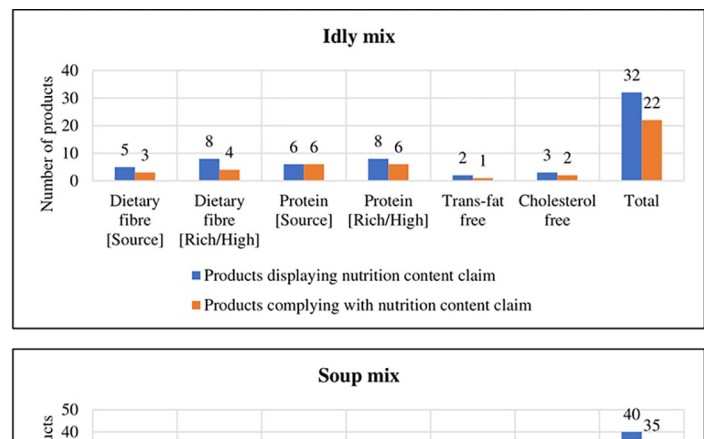

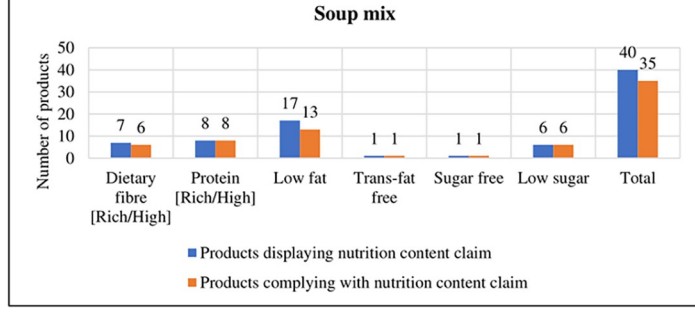

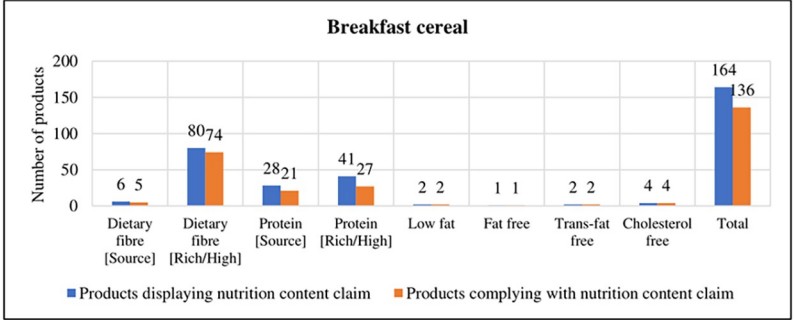

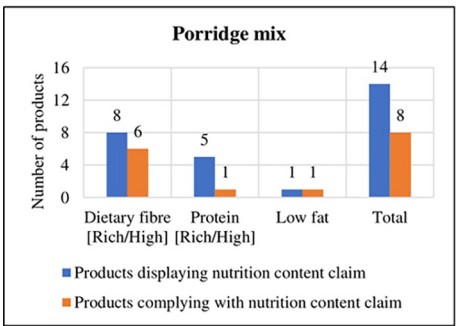
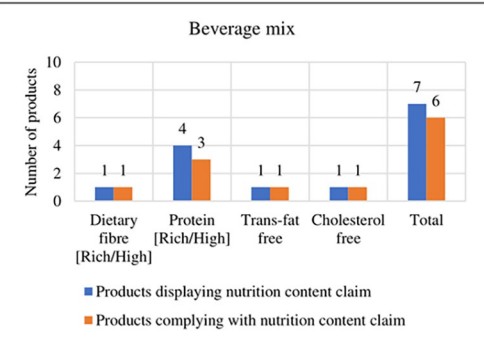

**Fig 3. Number of products displaying and complying with the nutrition content claims in the convenience food products and snacks.**

products in the extruded snacks food product category carried any claims. Low GI claim was present only in the 'beverage mix' (11%), 'idly mix' (4%), and, 'soup mix' (1.4%) categories. The percentage of products complied with the nutrition content claims in each category are as follows–porridge mix (57.1%), idly mix (68.8%), soup mix (87.5%), breakfast cereal (82.9%) and beverage mix (85.7%). Overall, 80.5% of convenience food products complied with the

nutrition content claims declared on the packs as per the mandate of FSSAI. Some of the products claiming to be high in certain nutrients such as protein or fibre, failed to meet FSSAI guidelines [16]. The non-compliance of the nutritional content claims was due to inaccurate claims and error/missing nutritional information which is required to verify the claims.

## Discussion

The present study explores the nutritional quality, labelling and nutritional claims displayed on packaged foods in the Indian market. The current study focused on the evaluation of nutrition labels declared on the foods to understand their quality. Our findings are essentially based on FOP and BOP information and the results show about 80.5% of products displaying the nutrition claims met the eligibility for the claims declared. The study indicated that all the products (except extruded snacks) provided >70% of the energy from carbohydrates, while extruded snacks provided >47% of the energy from fat. A clear labelling system will be a valuable tool to influence consumers to choose healthier food products.

FSSAI introduced new packaging and labelling regulations that require products to display the minimum nutritional information for energy, protein, carbohydrate, sugar, and total fat [10]. According to FSSAI, nutrition information should be given as "per 100 g" or "100 ml" or "per serving" of the product on the label. Among the selected food categories, only certain breakfast cereals and some beverages have widely disclosed the concept of per-serving information and a few have mentioned percentage RDA or % daily intake contribution of the products. The gaps observed in the declaration of nutritional information on food products were evaluated in the study. Furthermore, some products claimed to contain whole grains did not display them in the ingredient list and these claims may be misleading to the consumer. The present study also indicated the scarcity of low-GI food options and in the context of the growing prevalence of NCDs, the study underscores the need for such foods for the population.

Our recent study suggests the following percentage energy calorie recommendations from carbohydrates (56–60%), proteins (14–17%) and fat (20–24%) for the prevention of progression to T2D in normal glucose-tolerant subjects [19]. Slow-digesting carbohydrates are more suitable as they elicit a lower glycemic response, create lesser insulin demand and are helpful in glycemic control not only for the diabetic population but for all [20] and such foods are scarce in the market. In the market products, the energy provided by carbohydrates were higher; thus, the study depicts the energy contribution from carbohydrates to be higher in all the products except extruded snacks where the energy contribution from fat is higher (S1–S3 Figs). Such products are not desirable in a high-risk population and underscores the need for the development foods with higher energy contribution from proteins.

The main ingredients in malt-based beverages, corn and potato-based snacks, and rice and semolina-based *idli* mixes can contribute to higher glycemic properties. Our earlier studies have shown that *idli* prepared from either white rice or brown rice was of high GI category (Unpublished data) which is due to the typical food matrix (finer particle size of the rice and pulse constituent in the fermented batter and the fluffy texture attained after steaming). We observed a medium GI for *idli* prepared using sorghum grits (bigger particle size) and a higher proportion of pulse (black gram dhal) used for the preparation of *idli* [21]. Low GI convenience *idli* mixes based on whole grain cereals and pulses with higher protein content with superior sensory characteristics are essential not only for a population with diabetes but for all who are at high risk. Porridge mixes contain the finer particle size flour of the cereals and pulses used and also the method of cooking gelatinises the starch resulting in high GI [22]. Porridge mixes with low GI are essential; this can be achieved by including a higher proportion of pulses (to improve protein content and slow digesting carbohydrates), functional

ingredients such as soluble fibres, slow digesting carbohydrates, and resistant starches while formulating porridges. Many beverages were made with malt extract or malt-based carbohydrates which elevate the simple carbohydrate load [23]. Moreover, the majority of the beverage mixes were high in sugar, and two servings a day would exceed the sugar allowance for a balanced diet for adults which is 5g/portion according to the dietary guidelines of Indians [24].

Refined sugar is believed to contribute to insulin resistance, weight gain and metabolic syndrome by increasing glycaemic load and reducing satiety [25, 26]. Preparation of beverage mixes with higher protein content (with lower levels of branched-chain amino acids), soluble fibres with lower viscosity such as resistant maltodextrins, lower molecular weight fibres etc., with lower GI, superior micronutrient and sensory profile would help in improving the protein, fibre intake and expand healthy choices. Additionally, most of the soup mixes, and few products in 'extruded snacks', 'beverage mix' and '*idli* mix' categories were high in sodium and increased consumption of sodium has been linked to hypertension and worsening complications of diabetes [27] and there is a need for low sodium food choices.

The inclusion of more proportion of vegetables, the inclusion of fibre and protein sources, protein concentrates, and protein isolates to replace starchy ingredients, and reducing the sodium content in the formulation of soups would help to improve the protein, and fibre intake [28, 29] of the population. Breakfast cereals available in the market were predominantly cereal-based, and only a lower proportion of the products included pulses. The inclusion of pulses either in the flaked form or in the composite mix taken for extrusion (for extruded breakfast cereals) can improve the protein content but also provide slow-digesting carbohydrates and may help achieve a lower GI for the product. The preparation of more varieties of breakfast cereals with savoury flavours can also be explored to widen the breakfast cereal category of products. Snacks with lower fat and sodium content are essential, and several innovative food processing approaches are available to prepare snacks with lower fat content. Popping, puffing, and extrusion technologies could be utilised for the preparation of snacks with lower fat content. In this context, exploration of alternative technologies for preparation as well as coating of snacks with seasonings with lower fat content needs to be executed. In addition, the sensory acceptability of the product, process, product economics needs to be evaluated and matched with the market demand. Our findings are consistent with a study comparing the healthiness of packaged foods in different countries [30], in which Indian and Chinese packaged foods and beverages were ranked as least healthy, having the highest levels of saturated fat, total sugars and energy density. More products should be introduced keeping in mind the sedentary lifestyle and various NCDs [5]. Research on food product development should be emphasised towards developing nutritious options with consumer acceptability, safety, shelf life combined with nutrition education so that it can withstand the food market saturation.

The strength of the study is to the best of our knowledge is one of the most extensive studies on the evaluation of the nutritional quality and nutritional content claims of selected Indian food products. One of the limitations of the study is that it includes only foods available in Chennai, although we have included food products available pan India through an e-survey. Few nutritional information and claims may be missing in the study due to the low resolution of some of the FOP and BOP images in the websites. The data reported in the study were reliant only on the accuracy of the information provided on the FOP and BOP images provided in the websites. In addition, in the present study international standards were not considered for evaluating the food products, since the international guidelines didn't include standards for *idly mix* and porridge mix. Hence for the current study, we have considered only the national guidelines (FSSAI). Furthermore, in the current study salt content which is a part of traffic light system was not included for the categorization as in Indian products, mainly sodium

levels were declared rather than sodium chloride content. Sodium is contributed by not only sodium chloride but also food ingredients, several other food additives, preservatives making it difficult to compute the sodium chloride values.

Despite the aforementioned challenges, the traffic light system categorization of foods used in the study since a cross-sectional survey with a quasi-experimental design was conducted by National Institute of Nutrition, Hyderabad to examine the food label reading habits of the participants along with their views on the acceptability of various formats of front of pack nutritional label in India [31]. The results of the study concluded that the number of participants agreed the traffic light labelling helped them to identify healthy and unhealthy food was 85% and 83.4% respectively. Around 89.9% of participants declared the traffic light labelling is easier to understand. Hence, we have used traffic light system to categorize the food products.

## Conclusions

The present study provides a comprehensive evaluation of the nutrient profile of selected packaged foods marketed in India. The study also provided insights into the healthiness of packaged foods by nutritional profiling, compliance with nutrition and health claims and primary ingredients used in the preparation of convenience foods. Our analysis found the overall compliance of the nutrient content claims for the studied convenience foods is 80.5%, which is an indicator of better nutritional quality. The 'beverage mix' and 'extruded snacks' category was found to be high in sugar content (65.7%) and total fat content (96.7%) respectively and 'soup mix' was found to have higher amounts of sodium, which highlights the clear need for reformulating the products. The study also indicated the scarcity of low-GI foods suitable for the diabetic population. Though there were low GI options in the beverage, idly mix and soup mix categories, the rest of the product categories did not have low GI options. To conclude, it is essential to empower consumers to effectively interpret the nutritional information and claims on the packaging of convenience foods. In order to explore the nutritional value of packaged food products, it is important to repeat this research study on a regular basis at different locations in India to understand the quality of food products evolving with time.

## Supporting information

**S1 Fig. Mean energy (Calorie) distribution from carbohydrates, protein and fat.**
(TIF)

**S2 Fig. (a).** Energy contributed by the macronutrients in commercial products (Idly mix), **(b).** Energy contributed by the macronutrients in commercial products (Breakfast cereal), **(c).** Energy contributed by the macronutrients in commercial products (Porridge mix), **(d).** Energy contributed by the macronutrients in commercial products (Soup mix), **(e).** Energy contributed by the macronutrients in commercial products (Beverage mix), **(f).** Energy contributed by the macronutrients in commercial products (Extruded snacks).
(ZIP)

**S3 Fig. (a).** Carbohydrate content of the convenience food products and snacks, **(b).** Protein content of the convenience food products and snacks, **(c).** Total fat content of the convenience food products and snacks, **(d).** Saturated fat content of the convenience food products and snacks, **(e).** Trans fat content of the convenience food products and snacks, **(f).** Cholesterol content of the convenience food products and snacks, **(g).** Dietary fibre content of the convenience food products and snacks, **(h).** Sugar content of the convenience food products and snacks, **(i).** Sodium content of the convenience food products and snacks, **(j).** Energy

contributed by the convenience food products and snacks.
(ZIP)

**S1 Dataset.**
(XLSX)

## Acknowledgments

The authors acknowledge the guidance provided by Ms. Sudha Vasudevan, Senior Scientist and Head, Department of Foods Nutrition and Dietetics Research, Madras Diabetes Research Foundation for the study.

## Author Contributions

**Conceptualization:** Shanmugam Shobana.

**Data curation:** Shanmugam Shobana, Gopalakrishnan Sangavi.

**Formal analysis:** Gopalakrishnan Sangavi, Ramatu Wuni, Bakshi Priyanka.

**Funding acquisition:** Shanmugam Shobana, Karani Santhanakrishnan Vimaleswaran.

**Investigation:** Shanmugam Shobana, Bakshi Priyanka, Arun Leelavady, Dhanushkodi Kayalvizhi.

**Methodology:** Shanmugam Shobana, Bakshi Priyanka, Arun Leelavady, Dhanushkodi Kayalvizhi.

**Project administration:** Shanmugam Shobana.

**Supervision:** Shanmugam Shobana, Karani Santhanakrishnan Vimaleswaran, Viswanathan Mohan.

**Validation:** Shanmugam Shobana, Gopalakrishnan Sangavi.

**Visualization:** Gopalakrishnan Sangavi, Bakshi Priyanka.

**Writing – original draft:** Shanmugam Shobana, Bakshi Priyanka, Arun Leelavady.

**Writing – review & editing:** Shanmugam Shobana, Gopalakrishnan Sangavi, Ramatu Wuni, Ranjit Mohan Anjana, Kamala Krishnaswamy, Karani Santhanakrishnan Vimaleswaran, Viswanathan Mohan.

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
