## [Decision Letter · Decision Letter 0]

28 Oct 2024

PONE-D-24-46848Assessment of front and back of pack nutrition labels of selected convenience food products and snacks available in the Indian market

PLOS ONE

Dear Dr. Shanmugam,

Thank you for submitting your manuscript to PLOS ONE. After careful consideration, we feel that it has merit but does not fully meet PLOS ONE’s publication criteria as it currently stands. Therefore, we invite you to submit a revised version of the manuscript that addresses the points raised during the review process.

Kind regards,

António Raposo

Academic Editor

PLOS ONE

2. We note that your Data Availability Statement is currently as follows: [The data analyzed in the manuscript was collected from front and back of pack nutritional labels of various food products mentioned in the manuscript. The collected data was transcribed into a spreadsheet and used for analysis. All relevant data are within the manuscript and its Supporting Information files.]

Additional Editor Comments:

I invite the authors to revise the manuscript based on both reviewers' comments.

Reviewers' comments:

Reviewer's Responses to Questions

**Comments to the Author**

1. Is the manuscript technically sound, and do the data support the conclusions?

Reviewer #1: Yes

Reviewer #2: Yes

2. Has the statistical analysis been performed appropriately and rigorously? 

Reviewer #1: Yes

Reviewer #2: N/A

3. Have the authors made all data underlying the findings in their manuscript fully available?

Reviewer #1: Yes

Reviewer #2: Yes

4. Is the manuscript presented in an intelligible fashion and written in standard English?

Reviewer #1: Yes

Reviewer #2: Yes

5. Review Comments to the Author

Reviewer #1: Comments:

The manuscript addresses a valid research question within the journal's scope. Additionally, it explores an important issue regarding the nutritional information on the front and back of pack labels of packaged foods in the Indian market.

The title accurately reflects the manuscript's focus and is concise, while the abstract precisely describes the objectives, methods, results, and conclusions.

To improve the structure and flow of the manuscript, I would like to offer some comments, questions, and suggestions.

Abstract:

- The abbreviations FSSAI and INFORMAS should be defined upon their first mention.

Introduction:

- I suggest standardizing the _indentation_ of the paragraphs.

Methods in Data Analysis:

- I recommend removing the quotation marks around the terms, as it is clear that these were the terms used, making the quotation marks unnecessary.

Results:

Table 1a:

- Please include a legend below the table with definitions for SFA, RDA, and GI.

- In the legend, only the abbreviations can be used within the table.

- Include the information: Source: FSSAI, 2018, in the legend.

Table 1b:

- Include the information: Source: FSSAI, 2018, in the legend.

Line 111 (referring to Table 3):

- I suggest removing the quotation marks around the claims, as it is clear that these were the terms used, making the quotation marks unnecessary.

Table 3:

- Please include a legend below the table with a definition for GI.

Include the information: Source: FSSAI, 2018, in the legend.

Line 132 (referring to Fig. 2):

- I suggest removing the quotation marks around the claims, as it is clear that these were the terms used, making the quotation marks unnecessary.

Line 158 (referring to Fig. 3):

- I suggest removing the quotation marks around the claims, as it is clear that these were the terms used, making the quotation marks unnecessary.

Discussion:

- In the first paragraph, I suggest reformulating it to include only data from this research as a brief summary, before contextualizing in the following paragraphs.

Conclusion:

- I recommend including the percentage in the section between lines 274 and 280.

- I suggest removing the quotation marks around the claims, as it is clear that these were the terms used, making the quotation marks unnecessary.

Fig. 1:

- I suggest improving the image quality.

If abbreviations are used, I recommend providing them in the form of a legend, rather than combining definitions and abbreviations in the image.

Fig. 2:

- I suggest improving the image quality.

If abbreviations are used, I recommend providing them in the form of a legend, rather than combining definitions and abbreviations in the image.

Fig. 3:

- I suggest improving the image quality.

Reviewer #2: This study provides an interesting analysis of nutritional labeling on packaged foods in the Indian market, focusing on six categories of convenience foods and using regulatory frameworks and the nutritional traffic light system to assess healthiness.

The manuscript examines only six categories of packaged foods, which may not be representative of the broader range of convenience foods available in the Indian market. This restricts the generalizability of findings to all packaged foods and limits insight into other widely consumed food categories, such as snacks and sweets specific to regional diets.

Although the traffic light system is a well-recognized method, it has limitations, particularly in the Indian context where consumer understanding of this labeling system might be low. The study does not mention whether the system was validated for Indian consumers or if alternative methods that might be more culturally or regionally relevant were considered. Additionally, the traffic light system only focuses on fat, saturated fat, and sugar content, omitting other potentially relevant nutrients like sodium, which are significant in packaged foods and may affect overall health.

The paper does not clarify if nutritional content was analyzed per 100 grams or based on serving sizes. This can affect how “healthy” or “less healthy” a product is perceived, as nutrient density can vary widely depending on portion size. A more accurate assessment would consider both serving size and per 100-gram analysis to better reflect actual consumption patterns.

While the study suggests reformulation for extruded snacks to reduce fat content, it does not address the broader challenges of reformulation in the Indian food industry, such as taste preferences, preservation requirements, and cost implications. The recommendation could be further enhanced by discussing the feasibility of reformulation in alignment with market demands and cost-effectiveness for manufacturers, which is crucial for practical implementation.

The manuscript assesses product labels without considering consumer comprehension or response to nutritional information, which is essential to understand how effective labeling is in guiding healthier choices. Including consumer surveys or focus groups would add valuable insights into whether nutritional labels influence purchasing behavior or if additional educational interventions are necessary.

While the study references FSSAI (Food Safety and Standards Authority of India) regulations, it does not provide a comparison with international standards or guidelines, such as those from the World Health Organization (WHO) or other national regulatory bodies. This could provide perspective on how Indian regulations align with or diverge from global best practices and help in identifying potential areas for regulatory improvement.

TFinally, the paper finds an 80.5% compliance rate for nutrient content claims but does not explore why the remaining 19.5% did not comply. It would be helpful to understand whether non-compliance was due to inaccurate claims, misleading language, or other regulatory issues, as this could highlight areas for improved regulatory enforcement and consumer protection.

6. PLOS authors have the option to publish the peer review history of their article (what does this mean?). If published, this will include your full peer review and any attached files.

Reviewer #1: **Yes: **Marcela Gomes Reis

Reviewer #2: **Yes: **M. João Reis Lima

---

## [Author Response · Author response to Decision Letter 0]

13 Nov 2024

Dear Editor in Chief,

 We are submitting the revised manuscript PONE-D-24-46848 entitled ‘Assessment of front and back of pack nutrition labels of selected convenience food products and snacks available in the Indian market’. The authors are grateful to the reviewers for their insightful comments and critical assessment on the paper. The manuscript has been revised according to the suggestions of the reviewers. The queries of the reviewers have been addressed and the detailed response has been enclosed with the letter. As recommended, we have submitted the marked-up copy and unmarked version of the revised manuscript. We have revised the manuscript to meet the PLOS ONE journal’s style requirements and submitted the ‘minimal data set’ consisting of the data analyzed in the study. The response to reviewer comments has also been enclosed along with the revised manuscript.

---

## [Decision Letter · Decision Letter 1]

18 Nov 2024

Assessment of front and back of pack nutrition labels of selected convenience food products and snacks available in the Indian market

PONE-D-24-46848R1

Dear Dr. Shanmugam,

We’re pleased to inform you that your manuscript has been judged scientifically suitable for publication and will be formally accepted for publication once it meets all outstanding technical requirements.

Kind regards,

António Raposo

Academic Editor

PLOS ONE

Additional Editor Comments (optional):

Reviewers' comments:

Reviewer's Responses to Questions

**Comments to the Author**

1. If the authors have adequately addressed your comments raised in a previous round of review and you feel that this manuscript is now acceptable for publication, you may indicate that here to bypass the “Comments to the Author” section, enter your conflict of interest statement in the “Confidential to Editor” section, and submit your "Accept" recommendation.

Reviewer #1: All comments have been addressed

Reviewer #2: All comments have been addressed

2. Is the manuscript technically sound, and do the data support the conclusions?

Reviewer #1: Yes

Reviewer #2: Yes

3. Has the statistical analysis been performed appropriately and rigorously? 

Reviewer #1: Yes

Reviewer #2: Yes

4. Have the authors made all data underlying the findings in their manuscript fully available?

Reviewer #1: Yes

Reviewer #2: Yes

5. Is the manuscript presented in an intelligible fashion and written in standard English?

Reviewer #1: Yes

Reviewer #2: Yes

6. Review Comments to the Author

Reviewer #1: The authors have made changes to all the issues raised, I thank them for their comments and suggest that the manuscript be accepted.

Reviewer #2: Considering the modifications and comments made by the authors in the manuscript, I believe it can be published.

7. PLOS authors have the option to publish the peer review history of their article (what does this mean?). If published, this will include your full peer review and any attached files.

Reviewer #1: **Yes: **Marcela Gomes Reis

Reviewer #2: **Yes: **M João Reis Lima

---

## [Editor Report · Acceptance letter]

26 Nov 2024

PONE-D-24-46848R1 

PLOS ONE

Dear Dr. Shobana, 

I'm pleased to inform you that your manuscript has been deemed suitable for publication in PLOS ONE. Congratulations! Your manuscript is now being handed over to our production team.

Kind regards, 

on behalf of

Dr. António Raposo 

Academic Editor

PLOS ONE